# Preoperative overweight and obesity do not cause inferior outcomes following open-wedge high tibial osteotomy: A retrospective cohort study of 123 patients

Cheng-Yi Wu[1,2], Jen-Wei Huang[1], Chang-Hao Lin[1], Wei-Hsing Chih[1]*

1 Department of Orthopaedics, Ditmanson Medical Foundation Chia-Yi Christian Hospital, Chiayi, Taiwan,
2 Department of Nursing, Chung Jen Junior College of Nursing, Health Science and Management, Chiayi, Taiwan

* chihws@gmail.com

**Data Availability Statement:** All relevant data are within the paper and its Supporting Information files.

## Abstract

Open-wedge high tibial osteotomy (OWHTO) is effective in treating medial compartment osteoarthritis. The association between body mass index (BMI) and outcomes following OWHTO is being debated. This study compared radiographic and clinical outcomes between patients with preoperative overweight, obesity, and normal BMI following OWHTO for medial compartment osteoarthritis. In total, 123 patients (123 knees) who underwent OWHTO for medial compartment osteoarthritis were enrolled and were divided into normal-BMI (18.5−24.9 kg/m$^2$), overweight (25−29.9 kg/m$^2$), and obese (>30 kg/m$^2$) groups based on body mass index. The numeric rating scale for pain, mechanical tibiofemoral angle (mTFA), tibia tilting angle (TTA), and Western Ontario and McMaster Universities Osteoarthritis Index (WOMAC) for function were evaluated preoperatively and at postoperative follow-ups. The improvements of clinical and radiological outcomes in normal-BMI, overweight, and obese groups were not significantly different. The incidence of soft tissue irritation, wound infection, nonunion, and conversion to total knee arthroplasty were not significantly different between groups. The clinical and radiological outcomes in patients with preoperative overweight, obesity, and normal-BMI were not significantly different. Preoperative overweight and obesity thus has no effect on outcomes following OWHTO during the two years follow-up period. These findings cannot be generalized to patients with morbid obesity.

## Introduction

Good outcomes have been achieved with open-wedge high tibial osteotomy (OWHTO) for medial compartment osteoarthritis (OA) [1–13]. In a recent systematic review, studies on the influence of risk factors have revealed that old age, female sex, overweight, OA of higher severity, large preoperative varus angles, and alignment under correction might result in worse

**Funding:** The authors received no specific funding for this work.

**Competing interests:** The authors have declared that no competing interests exist.

outcomes. However, these risk factors that influence persistence of OWHTO outcomes are being debated [14].

An increase in body weight results in joint overload, which limits movements and increases joint stress, potentially increasing disability in terms of functional capacity and leading to increases in pain levels [15]. Given that the benefit of OWHTO lies in reducing the weight loading in the medial compartment of the knee joint [11, 13, 16–18], we speculated the larger loading on the knee joint in those obese patients might induce the inferior outcome. Overall, this study aimed to compare the radiographic and clinical outcomes of patients with preoperative overweight, obesity, and normal body mass index (BMI) following OWHTO for medial compartment OA.

## Materials and methods

This was a retrospective cohort study conducted in a single hospital, and all surgical procedures were performed by a senior surgeon. The study design was approved by our institutional review board (approval no. 2021082).

### Patient group

We enrolled consecutive patients from April 2004 to July 2019, and data were retrospectively collected from medical records. The inclusion criteria were as follows: symptomatic OA of the isolated medial compartment with genu varum, Kellgren and Lawrence (KL) grade 3 and 4 [19], underwent OWHTO, and follow-up of at least 2 years. Patients with any of the following criteria were excluded: traumatic knee arthritis, infectious disease history, underwent bilateral OWHTO, and spine problems.

The body weight and height of the patients were measured to calculate their preoperative BMI. On the basis of the calculated BMI, the cohort was divided into three subgroups according to the World Health Organization international classification of adults' body weight index. The normal-BMI group was composed of all patients with preoperative BMI between 18.5 and 24.9 kg/m$^2$. The overweight group was composed of all patients with preoperative BMI between 25 and 29.9 kg/m$^2$. The obese group as composed of all patients with preoperative BMI $\geq$ 30 kg/m$^2$.

### Operative technique

All surgeries were performed under general anesthesia, and the patients were placed on a radiolucent table, which allowed for the use of fluoroscopy to evaluate the patients from hip to ankle. We approached from the medial portion of the proximal tibia and packed one 4" × 4" sponge into the posterior cortex to protect the neurovascular structure. Under fluoroscopy, a 1.6-mm K-wire directed toward the upper end of the fibular head was used to mark the orientation of the cutting plane. Subsequently, the osteotomy was performed using an oscillating saw to cut along the guiding K-wire plane up to 1 cm from the lateral cortex. The lateral wall of the proximal tibia was kept intact to serve as a hinge. An anterior cortex osteotomy was then performed at a level above the tibia tuberosity. After the osteotomy, we increased the gap progressively by using a spreader to create an open wedge. The intraoperative knee alignment was then checked using fluoroscopy. The weight-bearing line was corrected to pass the Fujisawa point, which is 62.5% from the medial-tibial articular margin. The Fujisawa point matches the mechanical axis with 3–5˚ of valgus, and the required correction could be determined by draw a line from this point to the center of the femoral head and another line to the center of the ankle joint [20]. We measured the resultant gap and then harvested a tricortical triangular-shaped bone graft the same size as the gap from the ipsilateral iliac crest. To maintain the

original posterior tibial slope in the sagittal plane, care was taken to harvest the bone graft as a wedge-to-cuboid-like shape, with the outer cortex of the iliac crest as the opening distance, and to place the tricortical bone graft at the posterior one-third to one-fourth of the tibial gap. If the gap was >10 mm, the residual space was then filled with an artificial bone graft. The corrected alignment was fixed with a conventional T-plate (Synthes, Paoli, PA, USA) with the knee in a completely extended position. The wound was closed with a drain set to avoid hematoma formation.

## Postoperative care and rehabilitation

After discharge, each patient was followed up once every 2 weeks in the first month and once every month thereafter until 3 months after the junction healed. Follow-ups were then arranged at 6, 9, 12, and 24 months after the surgery. After the surgery, the knee was kept in an extended position for 2 weeks by using an extension knee brace. Isometric quadricep exercises followed by straight leg-raising exercises commenced after the surgery. The range of motion exercise began after 2 weeks. Partial weight-bearing, according to individual tolerability, was allowed 4 weeks after the operation, and full weight-bearing was achieved in 8–12 weeks.

## Outcome evaluation

Both clinical and radiographic outcomes were measured before and at preoperatively 1-year and 2-year after the surgery. During clinical examination, we evaluated pain severity by using the numeric rating scale (0–10). The knee function was assessed by the Western Ontario and McMaster Universities Osteoarthritis Index (WOMAC) score, and the score was converted to a 100-point scale from 0 (poorest rating) to 100 (best rating) [21]. A weight-bearing standing radiograph of the full length of the bilateral lower limbs was used for routine radiographic evaluation after bone healing, and the mechanical tibiofemoral angle (mTFA) and tibia tilting angle (TTA) were evaluated. The TTA was defined as the difference in the angles between the tibial articular line and anatomical line. All measurements were performed using digital X-ray software (PACS, Philips Easy Vision, Best, the Netherlands). Furthermore, donor site morbidity, soft tissue irritation, infection, nonunion, and conversion to total knee arthroplasty (TKA) were noted.

## Statistical analysis

The one-way analysis of variance was used to compare continuous variables between groups, and the chi-square test or Fisher's exact test was used to compare discrete variables. Analyses of repeated measures were performed using the generalized estimating equation method. All analyses were performed using IBM SPSS Statistics (Version 21.0. Armonk, NY, USA: IBM Corp.).

## Results

In total, 123 patients (123 knees) were eligible for the study according to the inclusion/exclusion criteria. Among these patients, 28 patients (28 knees) were included in the normal-BMI group, 59 patients (59 knees) were included in the overweight group, and 36 patients (36 knees) were included in the obese group. Baseline statistical analysis, including of age and sex, revealed no significant differences between the three groups (Table 1). For patients with normal-BMI, the proportion of KL grade 4 osteoarthritis is lower than overweight and obese patients.

**Table 1. Patient characteristics.**

| | Preoperative BMI | | | p-value |
|---|---|---|---|---|
| | **Normal** | **Overweight** | **Obese** | **p-value** |
| | **18.5–24.9** | **25.0–29.9** | **≥30** | |
| n. of patients | 28 | 59 | 36 | |
| BMI | 23.4±1.0 | 27.1±1.4 | 32.7±2.1 | <0.001 |
| Age | 59.8±6.4 | 60.2±6.3 | 58.6±7.4 | 0.523 |
| Male | 9(32.1%) | 16(27.1%) | 14(38.9%) | 0.489 |
| OA grade[a] | | | | |
| Grade III | 23(82.1%) | 41(69.5%) | 24(66.7%) | 0.363 |
| Grade IV | 5(17.9%) | 18(30.5%) | 12(33.3%) | |

[a] According to the Kellgren and Lawrence classification

Before surgery, the pain scores were similar between groups, and the WOMAC score for function was significantly lower in the obese group than the normal-BMI group (Table 2). In group comparison, the improvements of the pain score and the WOMAC score for function from baseline to the 1-year and 2-year follow-ups did not differ between groups.

The mTFA and TTA significantly improved from the preoperative to postoperative stage and lasted up to the 2-year follow-up in all groups (Table 3). In group comparison, the improvements of mTFA were not significant between groups at any time point. The improvement of TTA was significantly higher in the obese group at postoperative, and the improvements between groups were not significant at the 1-year and 2-year follow-up. The incidence of donor site morbidity, soft tissue irritation, wound infection, nonunion, and conversion to TKA were not significantly high in the overweight and obese groups (Table 4).

## Discussion

The most important finding of the present study is the improvement of pain and function in patients with preoperative overweight and obesity were equivalent to those in patients with normal BMI. Our results indicated adequate corrections in normal-BMI, overweight, and obese groups postoperatively, and their good alignment was maintained during the follow-up periods. This finding is consistent with recent reports of favorable radiographic results with minimal complications after HTO [22–26].

Obesity puts individuals at a mechanical disadvantage: greater muscle power is required to undertake activities, particularly lower limb tasks. Moreover, the effect of obesity on pain might change when performing various types of activities. During the single-leg stance in the gait cycle, a force of 3–6 times body weight is transmitted across the knee joint. These forces are increased several times during high-impact activities. Therefore, any increase in weight may be roughly multiplied by these factors to reveal the excess force across the knee while walking for an overweight person [15]. We used specific activities as outcomes to evaluate the role of high BMI in different situations. However, we found a lack of association between preoperative BMI and pain during various activities, specifically pain while walking on a flat surface, pain while going upstairs or downstairs, pain while standing upright, and pain while sitting or lying. The result indicated that patients with preoperative overweight and obesity did not have inferior results in terms of pain relief when performing daily activities after OWHTO.

A total of three patients experience wound infection during follow-up, and all of them were in the overweight group. However, the wound infection did not occur in the obese group,

**Table 2. Clinical outcomes.**

| | Preoperative BMI | | |
|---|---|---|---|
| | **Normal** | **Overweight** | **Obese** |
| | **18.5−24.9** | **25.0−29.9** | **≥30** |
| Pain score (0–10) | | | |
| While walking on flat surface | | | |
| Pre-operative | 6.4±1.7 | 6.0±1.9 | 6.4±2.1 |
| *p*-value | ref | 0.309 | 0.933 |
| 1-year follow-up | 1.4±1.5 | 1.4±1.4 | 1.5±1.5 |
| *p*-value | ref | 0.395 | 0.664 |
| 2-year follow-up | 1.8±2.2 | 1.1±1.4 | 1.5±1.9 |
| *p*-value | ref | 0.636 | 0.665 |
| While going up or down stairs | | | |
| Pre-operative | 6.9±2.1 | 7.4±1.6 | 7.3±1.9 |
| *p*-value | ref | 0.359 | 0.319 |
| 1-year follow-up | 1.3±1.7 | 1.4±1.7 | 1.5±1.6 |
| *p*-value | ref | 0.352 | 0.310 |
| 2-year follow-up | 1.8±2.1 | 1.1±1.7 | 1.6±1.9 |
| *p*-value | ref | 0.440 | 0.342 |
| While standing upright | | | |
| Pre-operative | 6.0±2.1 | 5.7±2.2 | 6.4±1.6 |
| *p*-value | ref | 0.574 | 0.387 |
| 1-year follow-up | 1.0±1.4 | 1.2±1.5 | 1.1±1.3 |
| *p*-value | ref | 0.312 | 0.560 |
| 2-year follow-up | 1.6±2.2 | 0.8±1.3 | 1.2±1.8 |
| *p*-value | ref | 0.456 | 0.211 |
| While sitting or lying | | | |
| Pre-operative | 4.1±1.8 | 4.3±2.0 | 4.7±2.1 |
| *p*-value | ref | 0.612 | 0.258 |
| 1-year follow-up | 0.6±1.0 | 0.8±1.3 | 0.5±0.9 |
| *p*-value | ref | 0.973 | 0.216 |
| 2-year follow-up | 1.0±1.8 | 0.4±1.0 | 0.5±1.3 |
| *p*-value | ref | 0.096 | 0.064 |
| WOMAC function (0–100) | | | |
| Pre-operative | 47.3±19.1 | 43.0±16.0 | 36.7±15.8 |
| *p*-value | ref | 0.298 | 0.017 |
| 1-year follow-up | 84.9±12.1 | 82.9±13.4 | 81.3±17.0 |
| *p*-value | ref | 0.627 | 0.273 |
| 2-year follow-up | 83.0±18.8 | 85.3±14.8 | 79.9±14.6 |
| *p*-value | ref | 0.204 | 0.118 |

indicated the association between infection and preoperative BMI cannot be drawn. Among these three patients with postoperative infection, the operation time was similar with other patients, and none of them had underlying diseases. According to our results, preoperative overweight and obesity are not risk factors for soft tissue irritation, infection, nonunion, and conversion to TKA.

In the present study, the bone graft from the iliac crest was used to fill the osteotomy gaps for decreasing the risk of nonunion, loss of correction, and lateral hinge fractures. The incidences of nonunion, loss of correction, and lateral hinge fractures were 0.8% (1/123), 0%, and

**Table 3. Radiographic outcomes.**

| | Preoperative BMI | | |
|---|---|---|---|
| | **Normal** | **Overweight** | **Obese** |
| | **18.5−24.9** | **25.0−29.9** | **≥30** |
| mTFA | | | |
| Pre-operative | 189.0±2.6 | 189.3±2.9 | 190.2±3.2 |
| p-value | ref | 0.583 | 0.086 |
| Post-operative | 176.0±2.7 | 176.9±2.6 | 176.3±2.5 |
| p-value | ref | 0.481 | 0.340 |
| 1-year follow-up | 178.5±3.4 | 178.5±3.5 | 179.1±3.7 |
| p-value | ref | 0.729 | 0.573 |
| 2-year follow-up | 177.3±3.4 | 178.3±3.9 | 178.7±4.1 |
| p-value | ref | 0.480 | 0.892 |
| TTA | | | |
| Pre-operative | 85.2±2.5 | 84.9±2.4 | 84.3±2.1 |
| p-value | ref | 0.662 | 0.110 |
| Post-operative | 95.6±2.7 | 95.3±2.9 | 96.4±3.0 |
| p-value | ref | 0.992 | 0.042 |
| 1-year follow-up | 95.5±2.5 | 94.9±2.9 | 95.7±3.3 |
| p-value | ref | 0.724 | 0.209 |
| 2-year follow-up | 95.1±3.1 | 94.9±3.0 | 94.7±3.7 |
| p-value | ref | 0.949 | 0.586 |

0%, respectively. A total of four (3.3%) patients suffered from transient donor site numbness, and the symptoms recovered within 3 months. We placed the bone graft closer to the posterior cortex, thus minimizing any significant change in posterior tibial slopes. The absolute change of posterior tibial slope between preoperation and postoperation was 1.8 ± 1.1 degree. According to our results, filling the osteotomy gaps with the bone graft from the iliac crest could be considered. About 40.7% (50/123) patients in our series were manual laborers, and most of patients had physical demands on the work and daily activities. Therefore, a strict postoperative protocol was aiming to ensure the full recovery of patients.

Our study has some limitations. First, we lacked records of BMI during the follow-up period. Thus, we could only assess the association between preoperative BMI and postoperative outcomes; the change in body weight during the follow-up period is a confounding factor. Second, a relatively small case number is also a limitation of the study. Interactions may occur between various risk factors, such as age and sex. However, evaluation using methods of

**Table 4. Complications.**

| | Preoperative BMI | | | |
|---|---|---|---|---|
| | **Normal** | **Overweight** | **Obese** | **p-value** |
| | **18.5−24.9** | **25.0−29.9** | **≥30** | |
| Donor site morbidity | 1(3.6%) | 3(5.1%) | 0(0.0%) | 0.448[a] |
| Soft tissue irritation | 1(3.6%) | 2(3.4%) | 1(2.8%) | 1.000[a] |
| Wound infection | 0(0.0%) | 3(5.1%) | 0(0.0%) | 0.319[a] |
| Nonunion | 0(0.0%) | 1(1.7%) | 0(0.0%) | 1.000[a] |
| Conversion to TKA | 0(0.0%) | 0(0.0%) | 0(0.0%) | 1.000[a] |

[a] Fisher's exact test

stratification and regression analysis was not appropriate in this study due to the relatively small case number. Third, we lacked information on the occupation of patients. Heavy labor can increase the loading on the knee joint, which might affect clinical outcomes and functional performance. Fourth, only seven patients had a BMI value of >35, restricting our ability to comprehensively assess the role of morbid obesity in outcomes. Fifth, the two years follow-up of patients is not adequately long to assess TKA conversion rate, a longer period of follow-up is necessary.

## Conclusion

In conclusion, the favorable radiologic result, pain relief, and functional recovery in patients with preoperative overweight and obesity were equivalent to those with normal BMI. Preoperative overweight and obesity do not cause inferior outcomes following OWHTO during the two years follow-up period. These findings cannot be generalized to patients with morbid obesity.

## Supporting information

**S1 Data. Study dataset.**
(XLSX)

## Acknowledgments

This manuscript was edited by Wallace Academic Editing.

## Author Contributions

**Conceptualization:** Cheng-Yi Wu, Wei-Hsing Chih.

**Formal analysis:** Chang-Hao Lin.

**Investigation:** Jen-Wei Huang, Chang-Hao Lin.

**Methodology:** Cheng-Yi Wu, Jen-Wei Huang, Chang-Hao Lin.

**Supervision:** Wei-Hsing Chih.

**Writing – original draft:** Cheng-Yi Wu.

**Writing – review & editing:** Wei-Hsing Chih.

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
