## [Decision Letter · Decision Letter 0]

26 Dec 2022

PONE-D-22-28050Preoperative overweight and obesity do not cause inferior outcomes following open-wedge high tibial osteotomy: a retrospective cohort study of 123 patientsPLOS ONE

Dear Dr. Chih,

Thank you for submitting your manuscript to PLOS ONE. After careful consideration, we feel that it has merit but does not fully meet PLOS ONE’s publication criteria as it currently stands. Therefore, we invite you to submit a revised version of the manuscript that addresses the points raised during the review process.

 It is an interesting study. However, both reviewers mention that the statistical analysis doubtful because of small groups. Therefore, the conclusions too, may not be convincing. Please, include all recommendations of both reviewers in your revised manuscript. 

We look forward to receiving your revised manuscript.

Kind regards,

Hans-Peter Simmen, M.D., Professor of Surgery

Academic Editor

PLOS ONE

Journal Requirements:

2. PLOS requires an ORCID iD for the corresponding author in Editorial Manager on papers submitted after December 6th, 2016. Please ensure that you have an ORCID iD and that it is validated in Editorial Manager. To do this, go to ‘Update my Information’ (in the upper left-hand corner of the main menu), and click on the Fetch/Validate link next to the ORCID field. This will take you to the ORCID site and allow you to create a new iD or authenticate a pre-existing iD in Editorial Manager. Please see the following video for instructions on linking an ORCID iD to your Editorial Manager account: https://www.youtube.com/watch?v=_xcclfuvtxQ.

5. PLOS requires an ORCID iD for the corresponding author in Editorial Manager on papers submitted after December 6th, 2016. Please ensure that you have an ORCID iD and that it is validated in Editorial Manager. To do this, go to ‘Update my Information’ (in the upper left-hand corner of the main menu), and click on the Fetch/Validate link next to the ORCID field. This will take you to the ORCID site and allow you to create a new iD or authenticate a pre-existing iD in Editorial Manager. Please see the following video for instructions on linking an ORCID iD to your Editorial Manager account: https://www.youtube.com/watch?v=_xcclfuvtxQ.

Reviewers' comments:

Reviewer's Responses to Questions

**Comments to the Author**

1. Is the manuscript technically sound, and do the data support the conclusions?

Reviewer #1: Yes

Reviewer #2: Partly

2. Has the statistical analysis been performed appropriately and rigorously? 

Reviewer #1: Yes

Reviewer #2: I Don't Know

3. Have the authors made all data underlying the findings in their manuscript fully available?

Reviewer #1: No

Reviewer #2: Yes

4. Is the manuscript presented in an intelligible fashion and written in standard English?

Reviewer #1: Yes

Reviewer #2: Yes

5. Review Comments to the Author

Reviewer #1: The aim of this study was to compare clinical outcomes after owHTO between three groups of patients: Normal weight (1), overweight (2) and obese patients (3). In 123 patients no difference was found in the three groups according to complications and outcome after two years of follow up.

Very clear paper with appropriate protocol for testing the study question. As discussed in the limitations section the authors did not have enough morbid obese patients to draw any conclusion about this patient population. This should be mentioned in the abstract- and paper-conclusion: WHO grading for adipose patients may help. A further recommendation is to add the follow up time to the conclusion section. To find TKA conversion rates a longer follow up would be needed for example.

The technique used by the authors with bone graft from the iliac crest is a surgical technique which is not widely used for owHTO due to up to 20% complications for iliac bone harvesting. In a prospective, randomized, controlled trial by Fucentese et al. 2019 routine use was not recommended. But maybe this technique has some advantages in adipose patients? It is an important point which should be elaborated in the discussion section. Further it would be interesting to know the complication rates of the bone harvesting site as well. Do you have these data?

The follow-up protocol is very strict and conservative compared to current literature. Should be discussed in the discussion section as well. Did you do slope measurements after the procedure and how do you perform with current literature with the strict extension for the first two weeks?

Very good limitations section.

Page 4 Line 86 ff: Further elaborate this important sentence. For readers not used to HTO this may be difficult to understand as it is.

Reviewer #2: The present study deals with the topic "Preoperative overweight and obesity do not cause inferior outcomes following open-wedge high tibial osteotomy: a retrospective cohort study of 123 patients". The high tibial osteotomy is experiencing a renaissance in the last decade, which is why the authors' topic is certainly interesting. Unfortunately, the statistical analysis in the results section does not go beyond a mere descriptive presentation of a very small basic population. This small basic population was then again divided into 3 even smaller study groups, which clearly impairs the statistical significance of the results. Therefore, it is not surprising that the authors could not show any significant differences between the groups in the evaluation of complications, although obesity is a known risk factor for surgical site infections. The fact that the postoperative weight course was not documented and evaluated accordingly also reduces the significance of the results of this study. This has already been discussed by the authors in the limitations of the study.

In the current form, the results of the study reflect the known conflict of cost-benefit trade-off between surgical indication for HTO and medical risks due to obesity, because due to the small number of patients a valid statement regarding the postoperative complication rate is not possible. Also, the gender of the patients and its influence on the outcome was not taken into account.

However, it is interesting to note that the radiological and clinical outcome parameters do not seem to differ in the different weight groups. Whether this is also an effect of the small groups should be tested by increasing the number of patients.

6. PLOS authors have the option to publish the peer review history of their article (what does this mean?). If published, this will include your full peer review and any attached files.

Reviewer #1: **Yes: **Samuel Haupt

Reviewer #2: No

---

## [Author Response · Author response to Decision Letter 0]

3 Jan 2023

We have uploaded the document of Response to Reviewers to the system.

Journal Requirements:

Response: 

We have adjusted our manuscript to meet PLOS ONE's style requirements. (P.1, Line 1−16; P.2, Line 18−19, 21−22, 24−25, 31−32, 35−36; P.3, Line 41, 56, 60; P.4, Line 75; P.5, Line 101; P.6, Line 110, 124; P.7, Line 130, Table 1 and Table 2; P.9, Table 3 and Table 4; P.10, Line 185; P.12, Line 238, 245, 248, References list; P.17, Line 370−371)

2. PLOS requires an ORCID iD for the corresponding author in Editorial Manager on papers submitted after December 6th, 2016.

Response:

We have link ORCID iD to the corresponding author’s PLOS profile.

3. In your Data Availability statement, you have not specified where the minimal data set underlying the results described in your manuscript can be found. PLOS defines a study's minimal data set as the underlying data used to reach the conclusions drawn in the manuscript and any additional data required to replicate the reported study findings in their entirety. All PLOS journals require that the minimal data set be made fully available.

Response:

We have included all relevant data in the Supporting Information file (S1_ data.XLSX). (P.17, Line 370−371)

4. We note that you have indicated that data from this study are available upon request. PLOS only allows data to be available upon request if there are legal or ethical restrictions on sharing data publicly.

Response:

Because we have included all relevant data in the Supporting Information file Data, our data availability statement should be updated as follows: All relevant data are within the manuscript and its Supporting Information files.

5. PLOS requires an ORCID iD for the corresponding author in Editorial Manager on papers submitted after December 6th, 2016.

Response:

We have link ORCID iD to the corresponding author’s PLOS profile.

Reviewer #1 Comments:

1. As discussed in the limitations section the authors did not have enough morbid obese patients to draw any conclusion about this patient population. This should be mentioned in the abstract- and paper-conclusion: WHO grading for adipose patients may help. 

Response:

Thanks for your suggestion. We have mentioned that the finding in the present study cannot be generalized to patients with morbid obesity.

We have made the corrections as follows:

These findings cannot be generalized to morbid obesity. (P.2, Line 39−40; P.12, Line 242−243)

2. A further recommendation is to add the follow up time to the conclusion section. 

Response:

Thanks for your suggestion. We have added the follow-up time to the conclusion section. (P.2, Line 38−39; P.12, Line 242)

3. To find TKA conversion rates a longer follow up would be needed for example.

Response:

Thanks for your suggestion. In our series, because the follow-up periods ranged from 2 to 15 years, we only used the two years follow-up data for all of these patients to keep more consistent. We have made the corrections in the limitation.

We have made the corrections as follows:

Fifth, the two years follow-up of patients is not adequately long to assess TKA conversion rate, a longer period of follow-up is necessary. (P.12, Line 235−237)

4. The technique used by the authors with bone graft from the iliac crest is a surgical technique which is not widely used for owHTO due to up to 20% complications for iliac bone harvesting. In a prospective, randomized, controlled trial by Fucentese et al. 2019 routine use was not recommended. But maybe this technique has some advantages in adipose patients? It is an important point which should be elaborated in the discussion section. Further it would be interesting to know the complication rates of the bone harvesting site as well. Do you have these data?

Response:

Thank you for raising an important question. To date, no study has reported the advantage of bone graft from the iliac crest in overweight and obese patients. We fill the osteotomy gaps with the bone graft from the iliac crest for decreasing the risk of nonunion, loss of correction, and lateral hinge fractures. In our series, the incidences of nonunion, loss of correction, and lateral hinge fractures were 0.8% (1/123), 0%, and 0%, respectively. A total of four (3.3%) patients suffered from transient donor site numbness and the symptoms recovered within 3 months. Among these four patients, one patient has a normal BMI, and the other three patients are overweight. According to our results, the incidence of donor site morbidity is acceptable, and the incidence of nonunion, loss of correction, and lateral hinge are low. Therefore, we think fill the osteotomy gaps with the bone graft from the iliac crest could be considered.

We have made the corrections as follows:

Furthermore, donor site morbidity, soft tissue irritation, infection, nonunion, and conversion to total knee arthroplasty (TKA) were noted. (P.6, Line 122−123)

The incidence of donor site morbidity, soft tissue irritation, wound infection, nonunion, and conversion to TKA were not significantly high in the overweight and obese groups (Table 4). (P.8−9, Line 174−176)

We added the incidence of donor site morbidity in Table 4. (P.9, Line 182)

In the present study, the bone graft from the iliac crest was used to fill the osteotomy gaps for decreasing the risk of nonunion, loss of correction, and lateral hinge fractures. The incidences of nonunion, loss of correction, and lateral hinge fractures are 0.8% (1/123), 0%, and 0%, respectively. A total of four (3.3%) patients suffered from transient donor site numbness, and the symptoms recovered within 3 months. We placed the bone graft closer to the posterior cortex, thus minimizing any significant change in posterior tibial slopes. The absolute change of posterior tibial slope between preoperation and postoperation was 1.8 ± 1.1 degree. According to our results, filling the osteotomy gaps with the bone graft from the iliac crest could be considered. (P.11, Line 213−222)

5. The follow-up protocol is very strict and conservative compared to current literature. Should be discussed in the discussion section as well. Did you do slope measurements after the procedure and how do you perform with current literature with the strict extension for the first two weeks? 

Response: 

Thank you for raising an important question. In the present study, 40.7% of patients were manual laborers, and most of patients had physical demands on the work and daily activities. Therefore, a strict postoperative protocol was aiming to ensure the full recovery of patients. An extension knee brace was used to keep the knee in an extended position. We placed the bone graft closer to the posterior cortex, thus minimizing any significant change in posterior tibial slopes. The absolute change of posterior tibial slope between preoperation and postoperation was 1.8 ± 1.1 degree.

We have made the corrections as follows:

After the surgery, the knee was kept in an extended position for 2 weeks by using an extension knee brace. (P.5, Line 105−106)

The absolute change of posterior tibial slope between preoperation and postoperation was 1.8 ± 1.1 degree. (P.11, Line 219−220)

In our series, 40.7% (50/123) of patients were manual laborers, and most of patients had physical demands on the work and daily activities. Therefore, a strict postoperative protocol was aiming to ensure the full recovery of patients. (P.11, Line 222−224)

6. Page 4 Line 86 ff: Further elaborate this important sentence. For readers not used to HTO this may be difficult to understand as it is.

Response:

Thanks for your suggestion. We have elaborated in detail about the Fujisawa point. We have made the corrections as follows: 

The weight-bearing line was corrected to pass the Fujisawa point, which is 62.5% from the medial-tibial articular margin. The Fujisawa point matches the mechanical axis with 3−5° of valgus, and the required correction could be determined by draw a line from this point to the center of the femoral head and another line to the center of the ankle joint. [20] (P. 5, Line 87−91)

Reviewer #2 Comments:

1. Unfortunately, the statistical analysis in the results section does not go beyond a mere descriptive presentation of a very small basic population. This small basic population was then again divided into 3 even smaller study groups, which clearly impairs the statistical significance of the results. Therefore, it is not surprising that the authors could not show any significant differences between the groups in the evaluation of complications, although obesity is a known risk factor for surgical site infections. The fact that the postoperative weight course was not documented and evaluated accordingly also reduces the significance of the results of this study. This has already been discussed by the authors in the limitations of the study. In the current form, the results of the study reflect the known conflict of cost-benefit trade-off between surgical indication for HTO and medical risks due to obesity, because due to the small number of patients a valid statement regarding the postoperative complication rate is not possible. Also, the gender of the patients and its influence on the outcome was not taken into account. However, it is interesting to note that the radiological and clinical outcome parameters do not seem to differ in the different weight groups. Whether this is also an effect of the small groups should be tested by increasing the number of patients.

Response:

Thanks for your suggestion. If the sample size is large enough, a matched design could also be a solution to assess the influence of interaction and confounding effects. In our results, the values of radiographic outcomes and pain score were really similar between groups, which decreasing a risk of resulting in false negative error. However, for comprehensively assessing the influence of risk factors, increasing sample size and performing the multivariable analysis is necessary.

---

## [Editor Report · Decision Letter 1]

6 Jan 2023

Preoperative overweight and obesity do not cause inferior outcomes following open-wedge high tibial osteotomy: a retrospective cohort study of 123 patients

PONE-D-22-28050R1

Dear Dr. Chih,

We’re pleased to inform you that your manuscript has been judged scientifically suitable for publication and will be formally accepted for publication once it meets all outstanding technical requirements.

Kind regards,

Hans-Peter Simmen, M.D., Professor of Surgery

Academic Editor

PLOS ONE
---

## [Editor Report · Acceptance letter]

11 Jan 2023

PONE-D-22-28050R1 

Preoperative overweight and obesity do not cause inferior outcomes following open-wedge high tibial osteotomy: a retrospective cohort study of 123 patients 

Dear Dr. Wei-Hsing Chih:

I'm pleased to inform you that your manuscript has been deemed suitable for publication in PLOS ONE. Congratulations! Your manuscript is now with our production department. 

Kind regards, 

on behalf of

Dr. Hans-Peter Simmen 

Academic Editor

PLOS ONE